

**Vertical profiles of cloud condensation nuclei number concentration**
**and its empirical estimate from aerosol optical properties over the**
**North China Plain**
Rui Zhang[1], Yuying Wang[1], Zhanqing Li[2,3], Zhibin Wang[4], Russell R. Dickerson[3], Xinrong Ren[3],
Hao He[3], Fei Wang[5], Ying Gao[6], Xi Chen[1], Jialu Xu[1], Yafang Cheng[7], Hang Su[8]
[1] Key Laboratory for Aerosol-Cloud-Precipitation of China Meteorological Administration, School
of Atmospheric Physics, Nanjing University of Information Science & Technology, Nanjing
210044, China
[2] State Key Laboratory of Remote Sensing Science, College of Global Change and Earth System
Science, Beijing Normal University, Beijing 100875, China
[3] Earth System Science Interdisciplinary Center, Department of Atmospheric and Oceanic Science,
University of Maryland, College Park, MD, USA
[4] Research Center for Air Pollution and Health, College of Environmental and Resource Science,
Zhejiang University, Hangzhou 310058, China
[5] Key Laboratory for Cloud Physics, Chinese Academy of Meteorological Sciences, Beijing,
100081, China
[6] School of Atmospheric Sciences, Nanjing University, Nanjing 210008, China
[7] Minerva Research Group, Max Planck Institute for Chemistry, 55128 Mainz, Germany
[8] Multiphase Chemistry Department, Max Planck Institute for Chemistry, 55128 Mainz, Germany
Correspondence to: Yuying Wang (yuyingwang@nuist.edu.cn)





## Abstract


To better understand the characteristics of aerosol activation ability and optical properties, a
comprehensive airborne campaign was implemented over the North China Plain (NCP) from May
8 to June 11, 2016. Vertical profiles of cloud condensation nuclei (CCN) number concentration
($N_{CCN}$) and aerosol optical properties were measured simultaneously. Seventy-two-hour air mass
back trajectories show that during the campaign the measurement region is mainly influenced by air
masses in northwest and southeast. Air mass sources, temperature structure, anthropogenic
emissions, and terrain distribution are factors influencing $N_{CCN}$ profiles. CCN spectra suggest that
the ability of aerosol activation into CCN is stronger in southeast air masses than in northwest air
masses and stronger in the free atmosphere than near the surface. Vertical distributions of aerosol
scattering Ångström exponent (SAE) indicate that aerosols near the surface mainly originate from
primary emissions consisting of more fine particles. The combined effect of aerosol lifting aloft and
long-distance transport increase SAE and make it vary more in the free troposphere than near the
surface. For parameterizing $N_{CCN}$, the equation $N_{CCN}=10^{\beta}\cdot\sigma^{\gamma}$ is used to fit the relationship between
$N_{CCN}$ and the aerosol scattering coefficient ($\sigma$) at 450 nm. The fitting parameters $\beta$ and $\gamma$ have linear
relationships with the SAE. Empirical estimates of $N_{CCN}$ at 0.7% water vapor supersaturation ($ss$)
from aerosol optical properties are thus retrieved for the two air
masses:    $N_{CCN}=10^{-0.22\cdot SAE+2.39}\cdot\sigma^{0.30\cdot SAE+0.29}$    for    northwest    air    masses    and
$N_{CCN}=10^{-0.07\cdot SAE+2.29}\cdot\sigma^{0.14\cdot SAE+0.28}$ for southeast air masses. The estimated $N_{CCN}$ at 0.7% $ss$ agrees
with that measured, although the performance differs between low and high concentrations in the
two air masses. The results highlight the important impact of aerosol sources on the empirical
estimate of $N_{CCN}$ from aerosol optical properties.

## 1. Introduction

Defined as the mixture of solid and liquid particles suspended in the air, aerosols have a great
impact on Earth's climate system via their direct and indirect effects (IPCC, 2021). They not only
alter Earth's radiation budget by absorbing and scattering solar radiation directly (e.g., Bond et al.,
2013) but also affect the radiation budget indirectly by serving as cloud condensation nuclei (CCN),



modifying the microphysical properties of clouds (e.g., Lohmann and Feichter, 2005; Andreae and
Rosenfeld, 2008). This is referred to as aerosol-cloud interactions (ACI). Many studies suggest that
good knowledge of the CCN activation ability is the key to quantitatively evaluating ACI and its
radiative forcing in models (e.g., Rosenfeld et al., 2014, 2016; Z. Li et al., 2016, 2019; Liu and Li.,
2020). However, this is uncertain because of the lack of comprehensive observations.

CCN is a subset of aerosols that can be activated at a certain water vapor supersaturation ($ss$).

The activation ability is mainly determined by three aerosol properties, namely, particle size,
chemical composition, and mixing state (e.g., Farmer et al., 2015; F. Zhang et al., 2017; Cai et al.,
2018; Y. Wang et al., 2018). Previous studies have reported that these three factors have large
spatiotemporal variabilities over different regions in the world (e.g., Juranyi et al., 2011; Paramonov
et al., 2015; Schmale et al., 2018), especially in fast-developing countries like China (Z. Li et al.,
2019). This increases the uncertainty of estimates of ACI.

To evaluate the effect of aerosols on air quality and atmospheric radiative forcing in China,

many field experiments have been carried out in recent years in some developed regions, such as
the Pearl River Delta (PRD) (e.g., Rose et al., 2010), the Yangtze River Delta (YRD) (e.g., Leng et
al., 2013), and the North China Plain (NCP) (e.g., L. J. Guo et al., 2015; F. Zhang et al., 2017; J.
Ren et al., 2018). Some of these studies including measurements of CCN aimed at investigating the
characteristics of CCN activation properties and their influential factors or establishing reasonable
estimation schemes for CCN number concentration ($N_{CCN}$). For example, L. J. Guo et al. (2015)
discussed the change in CCN activation properties in a long-lasting severe fog and haze episode. F.
Zhang et al. (2017) conducted $N_{CCN}$ closure experiments, finding that $N_{CCN}$ was well estimated using
the data of aerosol size number concentration and bulk chemical composition but it was influenced
by the aerosol aging level. J. Ren et al. (2018) suggested that it was better to predict $N_{CCN}$ using
aerosol size-resolved rather than bulk chemical composition data. However, most of these studies
were based on ground-based observations, which could not characterize the vertical distributions of
CCN properties and $N_{CCN}$ profiles. The CCN activation ability and $N_{CCN}$ below cloud bases are key
in quantifying ACI (Rosenfeld et al., 2014; Z. Li et al., 2016). Therefore, it is necessary to do more
studies about CCN profiles in China.

A commonly used platform to observe $N_{CCN}$ profiles and the vertical distribution of CCN

activation ability is an aircraft (e.g., J. Li et al., 2015b; Jayachandran et al., 2020a; Manoj et al.,





2021; Z. Cai et al., 2022). However, limited by high costs and technological complexity, current
aircraft measurements are insufficient to quantify ACI. Some studies have thus attempted to
estimate $N_{CCN}$ using aerosol optical data (e.g., Andreae, 2009; Liu and Li, 2014; Tao et al., 2018).
For example, Andreae (2009) built an exponential function between $N_{CCN}$ and aerosol optical depth
(AOD). Liu and Li (2014) defined the aerosol scattering index (AI) using aerosol scattering
coefficients ($\sigma$) and aerosol scattering Ångström exponent (SAE) to estimate $N_{CCN}$. Tao et al. (2018)
proposed a new method for estimating $N_{CCN}$ based on a three-wavelength humidified nephelometer
system. Most of these $N_{CCN}$ parameterization schemes, however, were conducted based on ground-
based observations in different regions and were rarely verified by in situ $N_{CCN}$ profiles.

Over the past few decades, rapid industrialization and urbanization have made the NCP one of

the most heavily polluted regions in China. The large number of aerosols and gases emitted by
human activities deteriorated air quality, strongly impacting the regional climate (e.g., Fan et al.,
2016; Chen et al., 2022). The aerosol activation ability and optical properties in the NCP have drawn
much attention (e.g., Zhang et al., 2016, 2017; Wang et al., 2018b). In light of this, we undertook a
comprehensive airborne campaign in the NCP under the aegis of a project called Air chemistry
Research In Asia (ARIAs). We directly measured profiles of $N_{CCN}$ and aerosol optical properties
from an aircraft and analyzed the CCN activation property and relationships between $N_{CCN}$ and
aerosol optical properties. This study will provide a perspective to improve aerosol-cloud
parameterizations applied in the NCP. Analytical methods developed here will also be applicable to
other regions of the world.

This paper is structured as follows. Details about the airborne campaign, instruments, and air

mass sources are given in Section 2. Section 3 discusses and analyzes $N_{CCN}$ profiles at 0.7% *ss*,
vertical distributions of CCN spectra, and possible relationships between $N_{CCN}$ and aerosol optical
properties. Section 4 summarizes the major conclusions of this study.

## 110   2. Airborne campaign, instruments, and air mass sources

2.1 Airborne campaign

Hebei province (36°05' N-42°40' N, 113°27' E-119°50' E) is located north of the Yellow River

and east of the Taihang Mountains in the NCP. It surrounds the Beijing and Tianjin megacities,





and borders Shangdong province to the east, Shanxi province to the west, Henan province to the
south, and the Inner Mongolia Autonomous Region to the north (Fig. 1a). The terrain of Hebei
province is high in the northwest and low in the southeast, with the altitude generally decreasing
from the northwest to the southeast. The plain area covers most of Hebei province, located in the
eastern foothills of the Taihang Mountains.

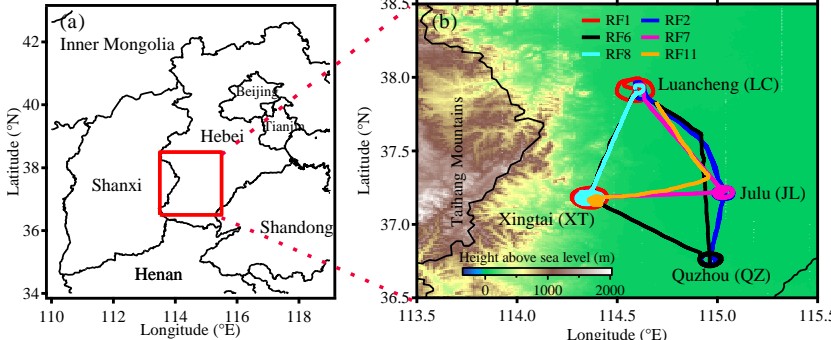

**Figure 1. (a)** The geographic location of Hebei province and **(b)** flight tracks of six flights
conducted over the southern plain of Hebei province in from May 8 to June 11, 2016. The colored
background shows terrain heights above sea level (unit: m).

The ARIAs campaign was carried out from May 8 to June 11, 2016 in the southern plain area

of Hebei province using a Y-12 turboprop airplane operated by the Weather Modification Office
of the Hebei Meteorological Bureau. The details of the flight plans were introduced in F. Wang et
al. (2018). Luancheng (LC, 114.36º E, 37.18º N; 182 m above sea level, or a.s.l.), Xingtai (XT,
114.36º E, 37.18º N; 182 m a.s.l.), Julu (JL, 115.02º E, 37.22º N; 20 m a.s.l.), and Quzhou (QZ,
114.96º E, 36.76º N; 40 m a.s.l.) are the four central sampling sites (Fig. 1b), all to the east of the
Taihang Mountains. Six flights (RF1, RF2, RF6, RF7, RF8, and RF11) measuring $N_{CCN}$ and aerosol
optical properties are used in this study. In all the flights, the Y-12 airplane conducted vertical
spiral flights from ~0.3 to ~3.5 km near one or two central sampling sites and level flights at
different fixed altitudes between different central sampling sites. Every flight obtained several
$N_{CCN}$ profiles at one or two sites and $N_{CCN}$ data at several fixed altitudes. Table 1 lists details about
the flight tracks (also see Fig. 1b).

Altitudes are distances a.s.l. in this study. All aircraft flights except RF8 (conducted from

16:30–18:24 CST; CST stands for China standard time, which is 8h ahead of UTC) were conducted



around noon (10:00–15:00 CST), when the planetary boundary layer (PBL) height was fully
developed.

**Table 1.** Detailed information about the flight tracks deployed during the campaign. Flight
code (third column): The number after 'RF' indicates the flight number, the number after '_'
indicates the number of vertical spiral flights, and the letter after '_' indicates the number of level
flights.

| Flight number, date | Time range (CST) | Flight code | Region covered | Vertical height a.s.l. (km) |
|---|---|---|---|---|
| RF1, 20160508 | 13:02–14:29 | RF1_1 | XT | 0.3–3.7 |
| | | RF1_a | track from XT to LC | ~3.6 |
| | | RF1_2 | LC | 0.3–3.2 |
| RF2, 20160515 | 12:17–15:04 | RF2_a | track from LC to JL | ~0.4 |
| | | RF2_1 | JL | 0.3–3.6 |
| | | RF2_2 | QZ | 0.3–3.6 |
| | | RF2_b | track from QZ to JL | ~3.6 |
| | | RF2_c | track from JL to LC | ~0.4 |
| RF6, 20160521 | 12:04–14:41 | RF6_1 | QZ | 0.3–3.1 |
| | | RF6_a | track from QZ to XT | ~2.5 |
| | | RF6_2 | XT | 0.3–2.6 |
| | | RF6_b | track from XT to LC | ~1.1 |
| RF7, 20160528 | 10:21–13:25 | RF7_a | track around XT | ~3.1 |
| | | RF7_1 | XT | 0.5–3.1 |
| | | RF7_b | track from XT to JL | ~0.4 |
| | | RF7_2 | JL | 0.3–2.5 |
| | | RF7_c | track from JL to LC | ~1.8 |
| RF8, 20160528 | 16:30–18:24 | RF8_a | track around XT | ~0.6 |
| | | RF8_1 | XT | 0.5–3.1 |
| RF11, 20160611 | 11:07–12:28 | RF11_a | track around XT | ~0.6 |
| | | RF11_1 | XT | 0.3–3.2 |


## 2.2 Instruments

To satisfy the needs of this study, the Y-12 airplane was equipped with a dual-column CCN
counter (CCNc), a three-wavelength integrating nephelometer, and a Cloud Water Inertial Probe
(CWIP). All instruments were calibrated rigorously prior to the airborne campaign. Table 2
summarizes the instruments equipped on the airplane.



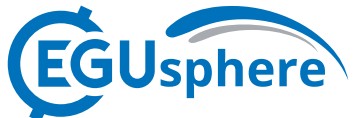

**Table 2.** Instruments equipped on the Y-12 airplane used in this study.

| Instrument | Parameter | Time resolution | Accuracy |
|---|---|---|---|
| CCN counter (model CCNc-200, DMT Inc.) | CCN number concentrations($N_{CCN}$) | 1 s | – |
| Nephelometer (model 3565, TSI Inc.) | Aerosol scattering coefficients ($\sigma$) at three wavelengths (450, 550, and 700 nm) | 1 s | 0.5 Mm$^{-1}$ |
| CWIP (Rain Dynamics Inc.) | Temperature ($T$) | 1 s | 1 K |
| | Relative humidity (RH) | 1 s | 2% |
| | Position | 0.1 s | – |


$N_{CCN}$ was measured by a dual-column continuous-flow thermal-gradient cloud condensation

nuclei counter (model CCN$_C$-200, DMT Inc.) with a time resolution of 1 s. It is equipped with two
columns that can simultaneously measure $N_{CCN}$ at two different *ss* levels without mutual
interference. In this campaign, only one *ss* level is set in the first column during all flights, but
eight different *ss* levels are set in the second column with a measurement time interval of 90 s for
each *ss* level. Considering the equilibrium time of *ss* levels, the final 30 s data at any *ss* level in the
cycle for the second column is used in this study. The *ss* level in columns was calibrated with pure
ammonium sulfate following procedures developed by Rose et al. (2008). The *ss* level in the first
column was corrected to 0.7% and the *ss* levels in the second column were corrected to 0.44%,
0.56%, 0.68%, 0.80%, 0.92%, 1.04%, 1.16%, and 1.28%. $N_{CCN}$ profiles at 0.7% *ss* and $N_{CCN}$ data
at different *ss* levels were thus available.

The integrating nephelometer (model 3565, TSI Inc.) can continuously measure aerosol

scattering coefficients ($\sigma$) at three wavelengths (450, 550, and 700 nm) with a time resolution of 1
s. Previous studies have shown that $\sigma$ becomes larger with increasing relative humidity (RH) due
to aerosol hygroscopic growth (e.g., L. Zhang et al., 2015; R. Ren et al., 2021). Hence, the RH of
sampled air was dried to below 40% in this campaign. The nephelometer was calibrated and tested
rigorously prior to the airborne campaign using carbon dioxide gas and filtered zero air. Anderson
and Ogren (1998) have provided details about the calibration methods and measurement
uncertainties of this nephelometer.

Ambient temperature ($T$) and RH were measured by a CWIP (Rain Dynamics Inc.) with a

time resolution of 1 s during flights. Real-time flight position data such as longitude, latitude, and



altitude were recorded by a global positioning system (GPS) and the CWIP with a time resolution
of 0.1 s. The CWIP time was calibrated and synchronized with the GPS time prior to deployment.

## 2.3 Air mass sources

Previous studies have suggested that differences in air masses will lead to spatiotemporal

differences in CCN activation ability and aerosol optical properties (e.g., Xu et al., 2020;
Jayachandran et al., 2020b). To better understand air mass sources and aerosol transport pathways
over the measurement area, seventy-two-hour air mass back trajectories for all $N_{CCN}$ profiles at 0.5,
1.5, 2.5, and 3.5 km are analyzed using the NOAA Hybrid Single Particle Lagrangian Integrated
Trajectory (HYSPLIT) model (Draxier and Hess, 1998). Results show that the sampling region is
mainly influenced by two distinct air masses, namely, northwest air masses and southeast air masses
(Fig. 2). Northwest air masses (Fig. 2a) originate from arid or semi-arid land, including five $N_{CCN}$
profiles whose flight codes are RF1_1, RF1_2, RF2_1, RF2_2, and RF11_1. Before these
trajectories approach the sampling area, most of these air masses flow around or are forced to lift
due to the influence of the Taihang Mountains. However, southeast air masses (Fig. 2b) originate
from coastal or marine areas, also including five $N_{CCN}$ profiles whose flight codes are RF6_1, RF6_2,
RF7_1, RF7_2, and RF8_1. Air masses in place during the RF7_1, RF7_2, and RF8_1 flights
originate from coastal areas, and those during the RF6_1 and RF6_2 flights originate from the
western Pacific. Southeast trajectories pass over the densely populated plain region to the east and
south of the sampling area, which is easily impacted by anthropogenic emissions. These trajectories
are also easily affected by differences in land and sea thermal properties, raising the air masses
gradually before reaching the sampling area (Fig. S1).





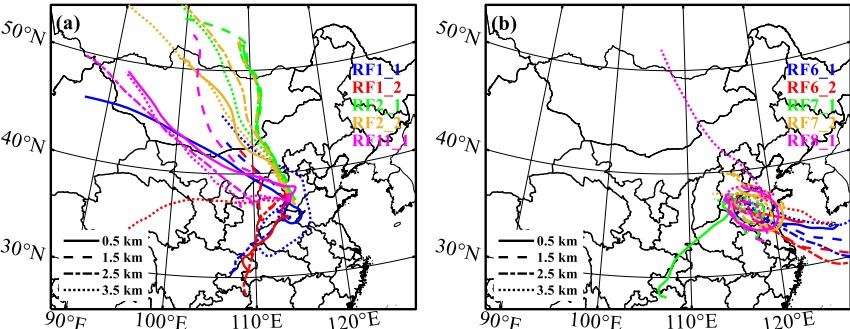


**Figure 2.** Seventy-two-hour HYSPLIT back trajectories over the sampling region: (**a**) northwest air
masses and (**b**) southeast air masses. The color of trajectories indicates different flight codes
associated with $N_{CCN}$ profiles. The line type shows trajectories with different starting altitudes (0.5,
1.5, 2.5, and 3.5 km).

**3. Results and Discussion**
3.1 Vertical distributions of $N_{CCN}$
3.1.1 Effect of the temperature inversion layer (TIL) on $N_{CCN}$ profiles
Previous studies have demonstrated the significant impact of the TIL structure on the vertical
distributions of aerosols and $N_{CCN}$ (e.g., Janhäll et al., 2006; J. Li et al., 2015a, 2015b). Here, $N_{CCN}$
profiles are classified into three categories according to the number of TILs (Table 3). Three typical
$N_{CCN}$ profiles at 0.7% *ss* (RF2_1, RF6_1, and RF1_1) with different numbers of TILs are chosen
for comparison purposes (Fig. 3; $N_{CCN}$ profiles associated with the other three flight codes are shown
in Figs. S2–4).

**Table 3.** Classification of different $N_{CCN}$ profiles based on the number of TILs.

| Categories | Flight codes of $N_{CCN}$ profiles |
|---|---|
| No TIL | RF2_1, RF2_2 |
| One TIL | RF6_1, RF6_2, RF7_1, RF7_2, RF8_1, RF11_1 |
| Two TILs | RF1_1, RF1_2 |


**No TIL:** Figure 3a shows vertical profiles of $T$ and potential temperature ($\theta$) for the RF2_1
$N_{CCN}$ profile (Fig. 3b). $T$ decreases with altitude in the absence of a TIL while the variation in $\theta$ with





altitude ($\partial\theta/\partial z$) is generally small below ~2.3 km (Fig. 3a). These meteorological conditions are
favorable for the upward transport of aerosols below ~2.3 km. The larger $\partial\theta/\partial z$ above ~2.3 km
suggests a more stable atmosphere, suppressing the upward transport of aerosols (Yau and Rogers,
1998). This is why $N_{CCN}$ peaks at ~2.3 km and decreases rapidly above (Fig. 3b). However, a second
$N_{CCN}$ peak is observed at ~3.2 km, with a small $\partial\theta/\partial z$ in the vicinity. The seventy-two-hour back
trajectory shows that the air mass in this case originates from the northwestern arid/semi-arid parts
of Mongolia (Fig. 2a). The long-distance transport of aerosols (like dust particles) may be
responsible for the $N_{CCN}$ peak at ~3.2 km. In another $N_{CCN}$ profile with no TIL (RF2_2), a weak
$N_{CCN}$ peak also appears at ~3.2 km (Fig. S2b). The RF11_1 $N_{CCN}$ profile with similar back
trajectories as RF2_1 and RF2_2 also has a weak $N_{CCN}$ peak at ~3.2 km. This suggests that the long-
distance transport of aerosols plays an important role in $N_{CCN}$ in the free troposphere over the NCP
under the influence of northwest air masses. Note that high $N_{CCN}$ in the free troposphere has an
important impact on cloud microphysical properties (Rosenfeld et al., 2008).
**One TIL:** The temperature profile in Fig. 3c shows a ~0.4-km-deep TIL at ~1.8 km. A thick
TIL hinders the upward transport of aerosols and facilitate the vertical mixing of $N_{CCN}$ below the
TIL. $N_{CCN}$ thus varies little with altitude below the TIL, with a mean $N_{CCN}$ at 0.7% $ss$ of 5140 cm$^{-3}$
(Fig. 3d). The $\theta$ profile in Fig. 3c suggests that $\partial\theta/\partial z$ above the TIL is much larger than below the
TIL, meaning a more stable atmosphere above the TIL. $N_{CCN}$ quickly decreases by an order of
magnitude from below to above the TIL (from 5542 cm$^{-3}$ at ~1.8 km to 365 cm$^{-3}$ at ~2.2 km).
Overall, the presence of a thick TIL has a large impact on the $N_{CCN}$ profile.
**Two TILs:** The temperature profile in Fig. 3e depicts two shallow TILs with the same depth
of ~0.2 km, appearing at ~0.8 km and ~2.5 km, respectively. Due to the hindering effect of a TIL on
the vertical transport of aerosols, only a small amount of CCN break through the first TIL and diffuse
to higher altitudes. Figure 3f suggests that $N_{CCN}$ increases with altitude from near the surface to the
bottom of the first TIL. A large amount of CCN accumulate below the first TIL, peaking at its bottom.
The second TIL makes $N_{CCN}$ accumulate again between the two TILs. Under the combined effect
of two TILs, the upward transport of CCN becomes difficult. The $\theta$ profile in Fig. 3e also shows
that $\partial\theta/\partial z$ is always positive, varying slightly with height. $N_{CCN}$ generally experiences a declining
trend with altitude between the two TILs (from 6380 cm$^{-3}$ at 0.9 km to 635 cm$^{-3}$ at 2.5 km). Above
the second TIL, $N_{CCN}$ remains at low and stable, with concentrations on the order of 10$^2$ cm$^{-3}$.




In summary, the TIL structure has an important impact on the vertical distribution of $N_{CCN}$.
Moreover, $N_{CCN}$ in the free troposphere are easily impacted by the long-distance transport of
aerosols under the influence of northwest air masses.

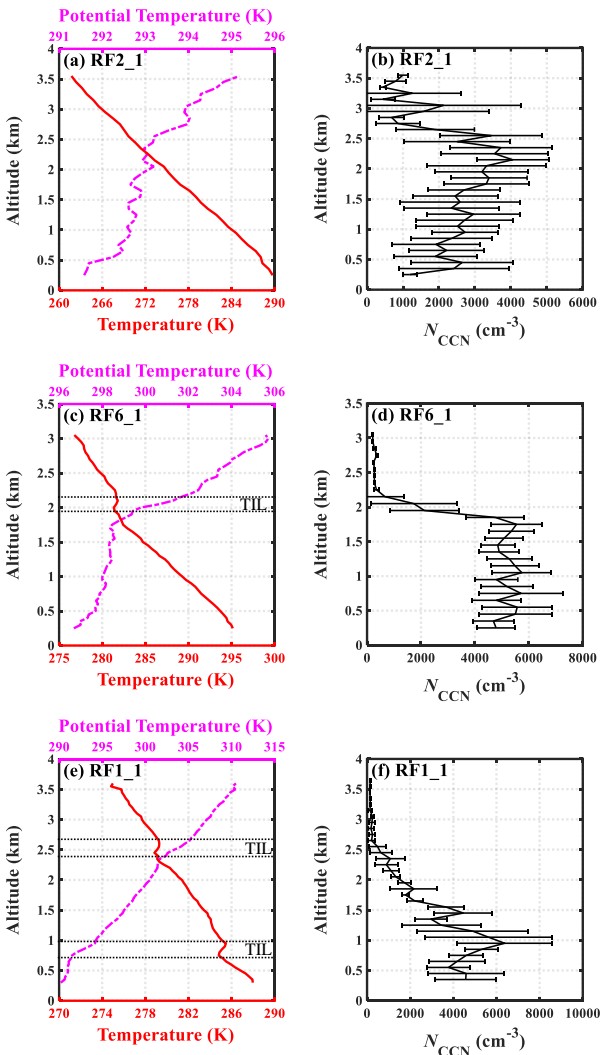


**Figure 3.** Vertical distributions of temperature ($T$) and potential temperature ($\theta$) (**a, c, e**), and $N_{CCN}$
at 0.7% $ss$ (**b, d, f**) for RF2_1, RF6_1, and RF1_1 $N_{CCN}$ profiles with (from top to bottom) no
temperature inversion layer (TIL), one TIL, and two TILs. Horizontal error bars represent standard
deviations of $N_{CCN}$ at 0.7% $ss$ at altitude intervals of 100 m.




3.1.2 Influence of air masses on $N_{CCN}$ profiles
To further investigate the influence of air masses on $N_{CCN}$ profiles, the mean $N_{CCN}$ at 0.7% $ss$
in different altitude ranges in two distinct air masses is analyzed (Fig. 4). In general, the mean $N_{CCN}$
at 0.7% $ss$ has a declining trend with increasing altitude in both air masses (Fig. 4a). The $N_{CCN}$ in
southeast air masses is higher than in northwest air masses below 1.5 km, indicating more aerosol
particles that can be activated as CCN in southeast air masses. Section 2.3 indicates that southeast
air masses always pass over the densely populated plain area. This means that massive
anthropogenic emissions can clearly increase $N_{CCN}$ near the surface. However, $N_{CCN}$ above 2 km is
much lower in southeast air masses than in northwest air masses. This further indicates that the long-
range transport of aerosols under the influence of northwest air masses contributes significantly to
$N_{CCN}$ in the free troposphere.
Figures 4b and 4c depict the mean $N_{CCN}$ at 0.7% $ss$ in different altitude ranges in northwest and
southeast air masses, respectively. Under the influence of northwest air masses, the temperature
structure varies, leading to different $N_{CCN}$ profiles (Fig. 4b). For RF2_1 and RF2_2 $N_{CCN}$ profiles
with no TIL (Figs. 3b and S2b), the combined effect of upward and long-distance transport of
aerosols increases $N_{CCN}$ at 0.7% $ss$ above 2 km. The $N_{CCN}$ from 2 to 2.5 km is even higher than near
the surface. For the RF11_1 $N_{CCN}$ profile with one TIL, $N_{CCN}$ at 0.7% $ss$ varies slightly with altitude.
For RF1_1 and RF1_2 $N_{CCN}$ profiles with two TILs, $N_{CCN}$ at 0.7% $ss$ above 2 km is much lower than
near the surface.
Under the influence of southeast air masses, the thermal structure for all $N_{CCN}$ profiles is similar,
with one TIL (Table 3). The $N_{CCN}$ profile patterns are thus similar, showing much lower $N_{CCN}$ above
2 km than near the surface (Fig. 4c). Figure 4c also suggests that $N_{CCN}$ at 0.7% $ss$ below 2 km is
higher in the RF6_1 and RF6_2 $N_{CCN}$ profiles than in the other three $N_{CCN}$ profiles (i.e., RF7_1,
RF7_2, and RF8_1). As discussed in section 2.3, air masses during RF6_1 and RF6_2 originate
from the western Pacific, while the others originate from coastal areas. This suggests that the impact
of marine aerosols is the possible reason for high $N_{CCN}$ in the RF6_1 and RF6_2 $N_{CCN}$ profiles.
Figure 4c also shows that the $N_{CCN}$ below 2 km is much higher at XT than at QZ and JL during the
same flights (RF6_2 vs. RF6_1, and RF7_1 vs. RF7_2). Figure 1b shows that the XT site is closer



to the Taihang Mountains than the QZ and JL sites. This implies that the terrain blocking effect of
the Taihang Mountains on aerosols accumulates aerosols, resulting in higher $N_{CCN}$ at XT.

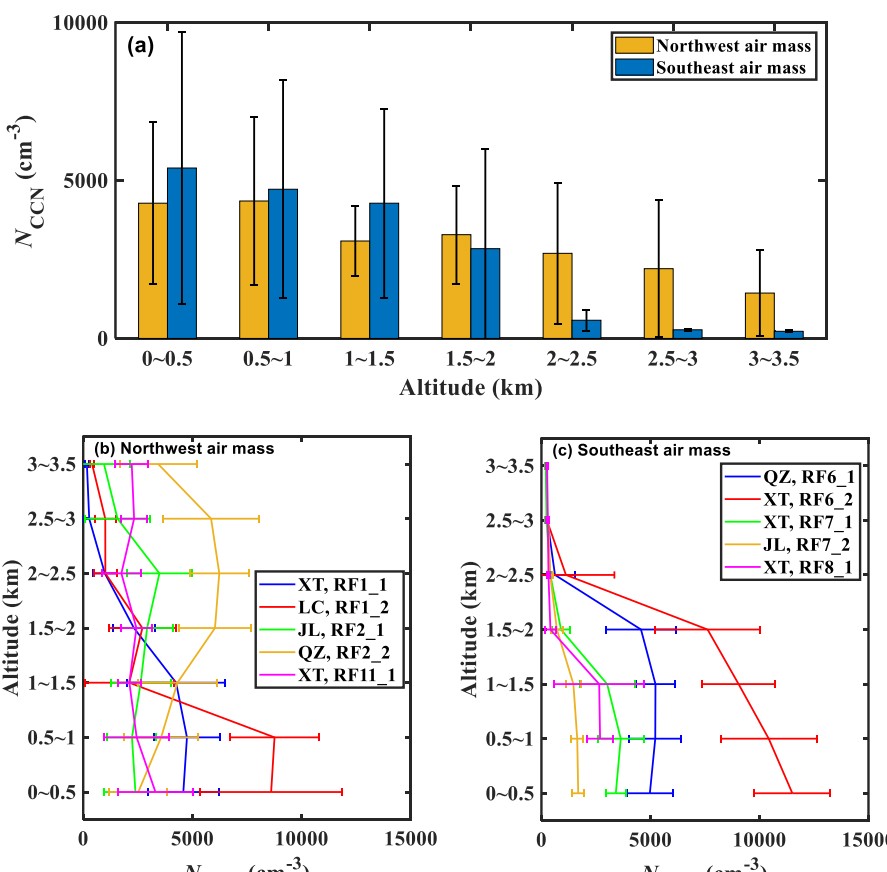

**Figure 4. (a)** Mean $N_{CCN}$ at 0.7% *ss* in different altitude ranges (ranging from 0 to 3.5 km at intervals
of 0.5 km) in northwest and southeast air masses, and for different $N_{CCN}$ profiles at 0.7% *ss* in **(b)**
northwest air masses and **(c)** southeast air masses. The different colors in (b) and (c) are for different
flights. Error bars represent standard deviations of $N_{CCN}$ at 0.7% *ss*.

In summary, $N_{CCN}$ profiles *ss* are influenced by multiple factors over the NCP. TIL structure,
aerosol long-range transport, and anthropogenic emissions lead to differences in the $N_{CCN}$ profiles
in different air masses. Even in the same air mass, diverse aerosol sources and terrain distributions
cause large differences in $N_{CCN}$.



## 3.2 Vertical distributions of CCN spectra in different air masses

The CCN spectrum is usually defined as a function of $N_{CCN}$ to $ss$. Twomey (1959) first reported that $N_{CCN}$ had an exponential relationship with $ss$. Since then, a variety of such functions have been proposed thanks to a large number of observations made which are all necessary given its nature of empirical relationships whose validity are generally limited. For example, Ji and Shaw (1998) provided a three-parameter function, while Gunthe et al. (2011) suggested a logarithmic function to fit CCN spectra. In this study, $N_{CCN}$ measurements made at different $ss$ during 11 level fights are used to fit CCN spectra. Twomey's relation (Twomey, 1959; Cohard et al., 1998) is used to fit the relationship between $N_{CCN}$ and $ss$ according to the least-squares method:

$$N_{CCN}(ss) = C \bullet (ss)^k \tag{1}$$

where $N_{CCN}(ss)$ is the $N_{CCN}$ at a specified $ss$, and $C$ and $k$ are two fitting coefficients. Table S1 lists the fitting results for the 11 level flights. In Eq. (1), the $C$ value represents $N_{CCN}$ at 1.0% $ss$, and the shape of the CCN spectrum is determined by the $k$ value. Previous studies have suggested that $k$ is closely related to the shape of particle number size distribution (PNSD) and aerosol hygroscopicity (e.g., Hegg et al., 1991; Jefferson, 2010). A lower $k$ value means a stronger aerosol activation ability (i.e., more coarse-mode particles or stronger aerosol hygroscopicity), and vice versa.

Figure 5 shows CCN spectra at different altitudes during three level flights (RF2, RF6, and RF7). The seventy-two-hour back trajectories (Fig. 2a) suggest that the RF2 flight is influenced by northwest air masses. The CCN spectra during three level flights (RF2_a, RF2_b, and RF2_c; Fig. 5a) shows that $C$ and $k$ are lower at 3.6 km (RF2_b) than at 0.4 km (RF2_a and RF2_c), indicating smaller $N_{CCN}$ but stronger aerosol activation ability in the free atmosphere than near the surface. At the same altitude (0.4 km), $C$ during the RF2_c flight (6560 cm$^{-3}$) is more than two times that during the RF2_a flight (3029 cm$^{-3}$), with different $k$ values (1.75 and 1.71, respectively). This indicates the regional variation of $N_{CCN}$ and the weak aerosol activation ability near the surface.

Figures 5b and 5c show CCN spectra during flights RF6 and RF7, which are influenced by southeast air masses (Fig. 2b). The $k$ values associated with southeast air masses (Fig. 5b and 5c) are always lower than those associated with northwest air masses (Fig. 5a). Therefore, aerosols in southeast air masses have a stronger activation ability than those in northwest air masses. This is likely because aerosols from southeast are mostly from anthropogenic emissions including more secondary particle matters such as sulfate and nitrate, while from northwest contains more natural



components such as mineral dust (Xia et al., 2019; Q. Wang et al., 2022). Figure 5c also shows that
$k$ during the RF7 flight decreases from 0.65 at 0.4 km to 0.37 at 1.8 km, increasing to 0.62 at 3.1
km. Figures S3c and S3e show that the altitude of the TIL during the RF7 flight is ~2 km. This
suggests that the aerosol activation ability near the TIL is stronger than that near the surface and in
the free atmosphere above the TIL. This implies that the hindering effect of the TIL promotes aerosol
aging processes, enhancing the aerosol activation ability (Y. Wang et al., 2018).

Overall, CCN spectra clearly varies with altitude over the NCP. The fitting coefficients of CCN

spectra ($C$ and $k$) are closely related to air mass sources, regional aerosol properties, and temperature
structure.

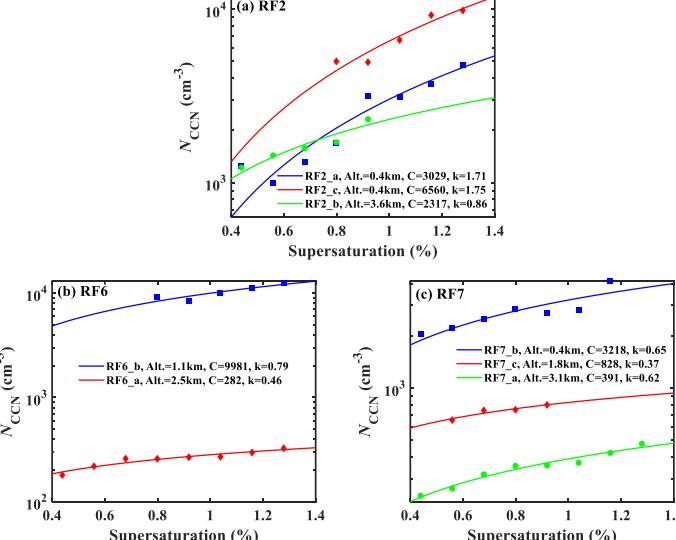

**Figure 5.** Fitted CCN spectra at different altitudes during three flights: (**a**) RF2, (**b**) RF6, and (**c**)
RF7. The flight code, flight altitude (Alt.), and the two fitting coefficients from Twomey's relation
($C$ and $k$) are given in each panel. Solid lines are the fitting lines described by Eq. (1).

## 3.3 The relationship between $N_{\mathrm{CCN}}$ and aerosol optical properties
### 3.3.1 Vertical distributions of aerosol scattering Ångström exponent (SAE)

The SAE is calculated as follows, where $\sigma(\lambda1)$ and $\sigma(\lambda2)$ are aerosol scattering coefficients at

two given wavelengths ($\lambda_1 = 450$ nm and $\lambda_2 = 700$ nm)



$$\text{SAE=-}\frac{\log\!\left(\sigma(\lambda_1)\right)\text{-}\log\!\left(\sigma(\lambda_2)\right)}{\log(\lambda_1)\text{-}\log(\lambda_2)} \qquad (2)$$

SAE is often used to qualitatively assess the dominant size mode of aerosols, reflecting the PNSD
pattern (e.g., Hamonou et al., 1999). A large SAE (> 2) generally implies that fine-mode aerosols
dominated (e.g., smoke particles), while a small SAE (< 1) means that the coarse mode dominated
(e.g., dust particles).
Figure 6a shows the vertical distributions of SAE during the vertical spiral flights. Three
profiles (RF2_1, RF2_2, and RF11_1) are not shown due to the lack of aerosol optical data. In
general, SAE decreases gradually with altitude, while its standard deviation increases with altitude.
This is likely because aerosols near the surface are easily influenced by primary emissions from
anthropogenic sources, consisting of more fine particles. The frequent appearance of a TIL at ~2 km
suppresses the upward transport of fine particles, leading to the rapid decrease of SAE above the
TIL. The long-distance transport of coarse-mode aerosols (like dust particles) also decreases SAE
in the free troposphere. As mentioned before, aerosol sources above 2 km are complex, which is
why the standard deviation of SAE is larger above ~2 km.
Figure 6b shows profiles of $N_{CCN}$ and $\sigma$ (data used here were collected at 0.7% $ss$ and 450 nm,
respectively) during the RF1_1 spiral flight. Figure S5 shows profiles from the other spiral flights.
In general, the vertical variation of $\sigma$ is synchronous with that of $N_{CCN}$, indicating that they are
correlated to some degree.

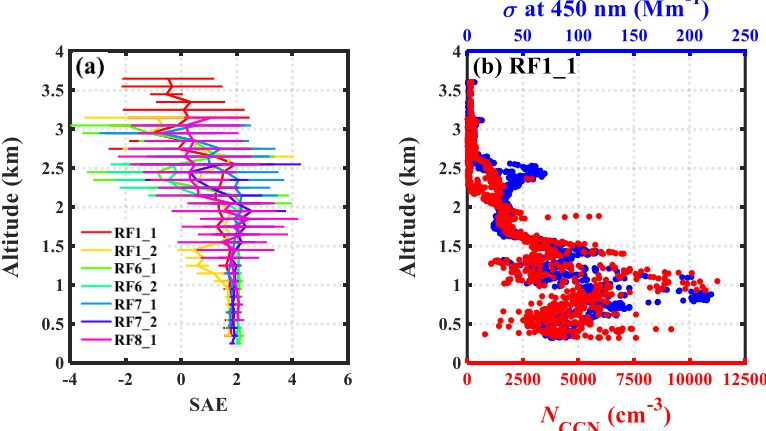


**Figure 6.** Vertical distributions of (**a**) aerosol scattering Ångström exponent (SAE) during the
vertical spiral flights (error bars are standard deviations of SAE) and (**b**) $N_{CCN}$ at 0.7% $ss$ (red dots)



and aerosol scattering coefficient ($\sigma$) at 450 nm (blue dots) during the RF1_1 vertical spiral flight.

3.3.2 Estimation of NCCN from aerosol optical properties

Both $N_{CCN}$ and aerosol optical properties are affected by the same factors (e.g., PNSD and

chemical composition). Therefore, numerous studies attempted to estimate $N_{CCN}$ for aerosol optical
properties, although there was no directly physical connection between them (e.g., Andreae, 2009;
Liu and Li, 2014; Tao et al., 2018). Previous studies indicated that the relationship between $N_{CCN}$
and $\sigma$ was non-linear, mainly due to the variation of PNSD patterns (e.g., Andreae, 2009; Shinozuka
et al., 2015). As discussed in section 3.3.1, SAE can be used to reflect the PNSD pattern. The clear
vertical variation of SAE (Fig. 6a) suggests a complex and variable relationship between $N_{CCN}$ at
0.7% $ss$ and $\sigma$ at 450 nm at different altitudes. Here, the parameterization provided by Shinozuka et
al. (2015) is used:

$$N_{CCN}=10^{\beta}\cdot\sigma^{\gamma} \tag{3}$$

where $\sigma$ is the aerosol scattering coefficient at 450 nm, and $\beta$ and $\gamma$ are two fitting parameters.
Shinozuka et al. (2015) suggested that $\beta$ and $\gamma$ were correlated to SAE, but the degree of correlation
differed in different regions. In this study, $N_{CCN}$ at 0.7% $ss$ and SAE data points are paired to derive
$\beta$ and $\gamma$. $N_{CCN}$ at other $ss$ levels are too little to do this work because of the loop measurement of
different $ss$ levels in the second column of CCNc-200.

Figure 7 shows the relationships between SAE and $\beta$ and SAE and $\gamma$ in two air masses. $\beta$ is

negatively correlated with SAE, while $\gamma$ is positively correlated with SAE. The correlations are
lower (smaller coefficients of determination, $R^2$) in northwest air masses than in southeast air masses,
likely due to more complex aerosol sources in northwest air masses. Empirical estimates of $N_{CCN}$ at
0.7% $ss$ from aerosol optical properties are determined as follows:

$$\text{Northwest air mass: } N_{CCN}=10^{-0.22\cdot SAE+2.39}\cdot\sigma^{0.30\cdot SAE+0.29} \tag{4}$$

$$\text{Southeast air mass: } N_{CCN}=10^{-0.07\cdot SAE+2.29}\cdot\sigma^{0.14\cdot SAE+0.28} \tag{5}$$




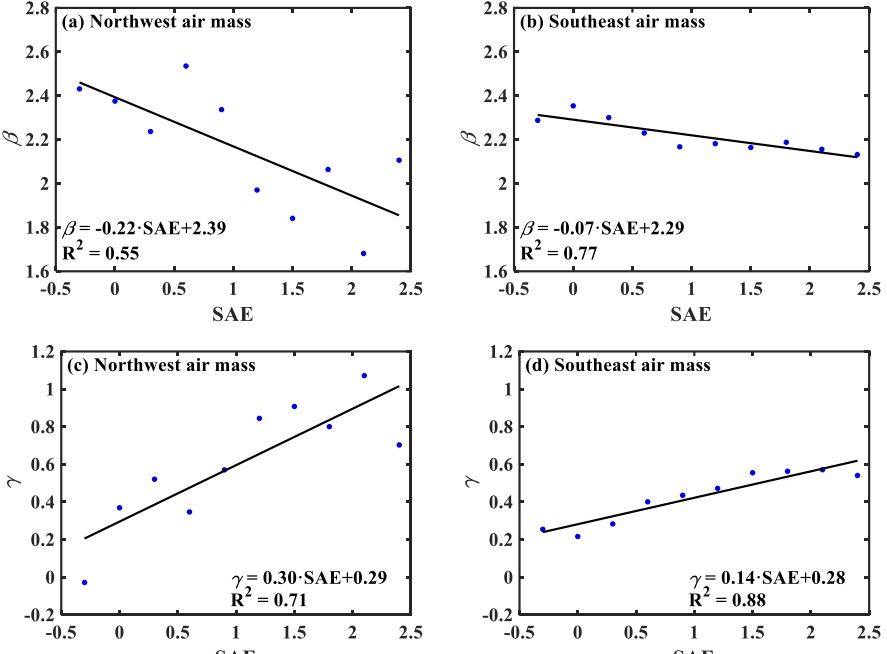

**Figure 7.** The two fitting parameters $\beta$ and $\gamma$ as a function of the aerosol scattering Ångström exponent (SAE) in northwest air masses (**a** and **c**) and southeast air masses (**b** and **d**). The dots are mean values averaged in 0.3-wide SAE bins. The black lines are best-fit lines from linear regression. Linear relations and coefficients of determination are given in each panel.

Figure 8 shows the comparisons of measured $N_{CCN}$ at 0.7% $ss$ and estimated $N_{CCN}$ at 0.7% $ss$ using Eqs. (4) and (5) for different vertical spiral flights in northwest and southwest air masses. For both air masses, most points approach the 1:1 line, indicating reasonable estimates using Eq. (4) and (5) to parameterize $N_{CCN}$. For northwest air masses (Fig. 8a), $N_{CCN}$ estimates are better under high concentration conditions than under low concentration conditions. However, for southeast air masses, $N_{CCN}$ estimates are better under low concentration conditions than under high concentration conditions. This is likely related to various aerosol sources at different altitudes. As previously discussed, most low $N_{CCN}$ values are observed in the upper atmosphere above the TIL, while high $N_{CCN}$ values are observed below the TIL. In northwest air masses, aerosol sources in the upper atmosphere are diverse, including the upward and long-distance transport of aerosols. This is why $N_{CCN}$ estimates worsen under low $N_{CCN}$ conditions. In southeast air masses, a single but thick TIL makes most aerosols accumulate in the lower atmosphere, where local emissions and the impact of




marine aerosols exacerbate $N_{CCN}$ estimates. These results highlight the important impact of aerosol
sources on the empirical estimate of $N_{CCN}$ from aerosol optical properties.

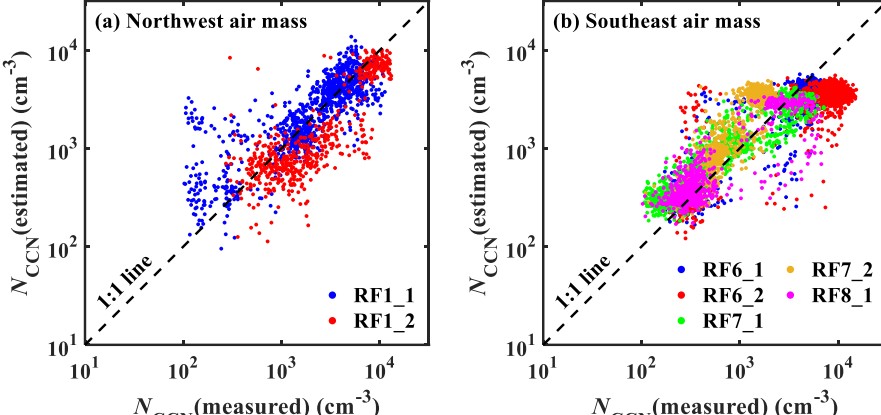


**Figure 8.** Comparisons between measured $N_{CCN}$ at 0.7% *ss* and estimated $N_{CCN}$ at 0.7% *ss* using
Eqs. (4) and (5) for different vertical spiral flights in (**a**) northwest and (**b**) southeast air masses.

## 4. Conclusions

A comprehensive airborne campaign was conducted over the North China Plain (NCP) under
the aegis of a project called Air chemistry Research In Asia (ARIAs). Seventy-two-hour air mass
back trajectories show that the region of study during this campaign is mainly influenced by
northwest and southeast air masses, originating from arid/semi-arid regions and coastal or marine
areas, respectively. In this study, the profiles of cloud condensation nuclei number concentration
($N_{CCN}$) and their estimates from aerosol optical properties are analyzed.
It is found that $N_{CCN}$ profiles at the water vapor supersaturation (*ss*) of 0.7% are impacted
largely by the temperature structure in the atmosphere. In general, the presence of a temperature
inversion layer (TIL) suppresses the upward transport of aerosols from near the surface, which is
affected by the number and thickness of TILs. In addition, air mass sources have a significant impact
on $N_{CCN}$ profile characteristics. Under the influence of northwest air masses, $N_{CCN}$ in the free
troposphere are easily impacted by the long-distance transport of aerosols. However, under the
influence of southeast air masses, atmospheric thermal structures for all $N_{CCN}$ profiles are similar,





with one TIL present in all cases. The patterns of $N_{\mathrm{CCN}}$ profiles are also similar, showing much lower
$N_{\mathrm{CCN}}$ above the TIL than near the surface. In addition to the impact of anthropogenic emissions, the
transport of marine aerosols is another reason for the high $N_{\mathrm{CCN}}$ near the surface when a southeast
air mass is present. Moreover, comparisons of $N_{\mathrm{CCN}}$ profiles during the same flights suggests that
the terrain blocking effect of the Taihang Mountains on aerosols accumulates aerosols, resulting in
high $N_{\mathrm{CCN}}$ near the mountains.

The Twomey's relation ($N_{\mathrm{CCN}}(ss)=C{\bullet}(ss)^{k}$, where $C$ and $k$ are two fitting coefficients) is used

to analyze CCN spectra and aerosol activation ability in this study. In general, there is a clear change
in CCN spectra with altitude. The aerosol activation ability in southeast air masses is stronger than
in northwest air masses, mainly due to the different chemical composition associated with diverse
air masses. In addition, the aerosol activation ability is stronger in the free atmosphere than near the
surface. The hindering effect of a TIL on the upward transport of aerosols promotes aerosol aging
processes, enhancing the aerosol activation ability near the TIL. The vertical distribution of aerosol
scattering Ångström exponent (SAE) indicates that aerosols near the surface are easily influenced
by primary emissions, consisting of more fine particles. The combined effect of aerosol upward and
long-distance transport increases SAE and make it vary more in the free troposphere than near the
surface.

The comparison of $N_{\mathrm{CCN}}$ at 0.7% $ss$ and aerosol scattering coefficient ($\sigma$) at 450 nm suggests

that the vertical variation of $\sigma$ is synchronous with that of $N_{\mathrm{CCN}}$. The equation, $N_{\mathrm{CCN}}=10^{\beta}{\cdot}\sigma^{\gamma}$ ($\beta$ and
$\gamma$ are two fitting parameters), is used to parameterize $N_{\mathrm{CCN}}$, with the parameters $\beta$ and $\gamma$ being linearly
correlated with the SAE. Empirical estimates of $N_{\mathrm{CCN}}$ at 0.7% $ss$ from aerosol optical properties are
thus    retrieved    (    $N_{\mathrm{CCN}}=10^{-0.22{\cdot}\mathrm{SAE}+2.39}{\cdot}\sigma^{0.30{\cdot}\mathrm{SAE}+0.29}$    for    northwest    air    masses,    and
$N_{\mathrm{CCN}}=10^{-0.07{\cdot}\mathrm{SAE}+2.29}{\cdot}\sigma^{0.14{\cdot}\mathrm{SAE}+0.28}$ for southeast air masses). The closure between the estimated and
measured $N_{\mathrm{CCN}}$ at 0.7% $ss$ is acceptable although different performances are seen under low and
high concentration conditions for the two air masses. Results suggest the important impact of aerosol
sources on the empirical estimate of $N_{\mathrm{CCN}}$ from aerosol optical properties.

$N_{\mathrm{CCN}}$ profiles in the NCP are impacted by multiple factors, including temperature structure,

air mass sources, anthropogenic emissions, and terrain distribution. These factors make estimating
$N_{\mathrm{CCN}}$ from aerosol optical properties more difficult. In the future, more aircraft measurement data
will be needed to establish a more reasonable parameterization scheme for $N_{\mathrm{CCN}}$ at different $ss$. This



study may also be useful for studying aerosol activation ability in other regions of the world.

*Acknowledgements.* This work was funded by the National Natural Science Foundation of China
(NSFC) research project (grant nos. 42005067 and 42030606), the National Science Foundation of
the United States (grant no. 1558259). We also thank all participants in the campaign for their
tireless work and cooperation, especially the work of the Hebei Weather Modification Office.

*Data availability.* Measurement data from the field campaign used in this study are available from
the corresponding author upon request (yuyingwang@nuist.edu.cn).

*Author contributions.* ZL and YW determined the main goal of this study. RZ and YW conceived
the study and prepared this paper. ZL, RD, HS, and YC led the airborne campaign, ZW, XR, HH,
and FW conducted this airborne campaign. HS, YC, and ZW provided the CCN data. YG, XC, and
JX processed the measurement data. All co-authors participated in science discussions and
suggested analyses.

*Competing interests.* The authors declare that they have no conflict of interest.

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
