# Peer review of "Vertical profiles of cloud condensation nuclei number concentration"

_EGUsphere, 2022_

## Author Comment (AC1)

**Reply to RC1**

We thank the reviewer for providing insightful comments and helpful suggestions that have substantially improved the manuscript. Below we have included the review comments in black followed by our responses in blue. In the revision of this manuscript, we have highlighted those changes accordingly.

This manuscript presents the airborne cloud condensation nuclei (CCN) measurements taken during the ARIAs (Air chemistry Research In Asia) campaign. The authors use HYSPLIT trajectories to identify the source regions of different air masses measured during the campaign, and present the results separately for air masses coming from two main directions (northwest and southeast). They show the impact of atmospheric stability on the vertical distribution of CCN. Furthermore, they parametrize the number concentration of CCN (Nccn) in terms of aerosol optical properties. The manuscript presents a novel height resolved in-situ Nccn data and has good potential for publication in ACP only after implementing and addressing the following comments.

Lines 90-92, 'Tao et al. (2018) proposed … system'. I don't understand how this is related to the idea of this paragraph. Did they give any empirical relationship between Nccn and optical properties? If yes, then it should be stated.

RE: Yes, they gave the empirical relationship. This sentence is revised as: "Tao et al. (2018) established a lookup table that includes $\sigma$, hygroscopicity parameter ($\kappa$), and Ångström exponent ($\mathring{A}$) for estimating $N_{CCN}$ based on the measurement of a three-wavelength humidified nephelometer system.".

Lines 92-93, 'Most of these… in situ Nccn profiles'. In atmospheric remote sensing, the word "profile" usually refers to a vertical representation. The parametrization schemes are mostly focused on estimating Nccn at ground. So there's no way one can compare/validate them with Nccn "profiles". I suggest replacing the word. Overall, I found the fourth paragraph of introduction to be confusing and suggest to modify it. It starts with the in situ Nccn "profile" measurements and the challenges involved in it. Thereafter how researchers have come up with empirical relations to estimate Nccn at "ground" using column integrated aerosol optical properties (AOD, AI, SAE). The ending sentence again discuss the how there's no validation with in situ Nccn "profile".

RE: That's a good suggestion. The fourth paragraph is revised as shown in the below.

"A commonly used platform to observe vertical distributions of $N_{CCN}$ and CCN activation ability is an aircraft (e.g., J. Li et al., 2015b; Jayachandran et al., 2020a; Manoj et al., 2021; Z. Cai et al., 2022). However, limited by high costs and technological complexity, current aircraft measurements are insufficient to quantify ACI. Some studies have thus attempted to estimate $N_{CCN}$ using aerosol optical data that are much more plentiful (e.g., Andreae, 2009; Liu and Li, 2014; Tao et al., 2018). For example, Andreae (2009) built an exponential function between $N_{CCN}$ and aerosol optical depth (AOD). Liu and Li (2014) found that the relationship between $N_{CCN}$ and AOD becomes invalid when the relative humidity (RH) exceeds 75% and they developed new parameterized relationships to estimate $N_{CCN}$ accounting for RH, particle size, and single scattering albedo (SSA). Tao et al. (2018) established a lookup table that includes $\sigma$, hygroscopicity parameter ($\kappa$), and Ångström exponent ($\mathring{A}$) for estimating $N_{CCN}$ based on the measurement of a three-wavelength

humidified nephelometer system. The vertical distributions of $N_{CCN}$ were also estimated using lidar data. Lv et al. (2018) developed an algorithm for profiling $N_{CCN}$ using backscatter coefficients at 355, 532, and 1,064 nm and extinction coefficients at 355 and 532 nm from multiwavelength lidar systems. Satellite lidar data of the Cloud–Aerosol Lidar and Infrared Pathfinder Satellite Observations (CALIPSO) have also been employed to retrieve the profiles of lidar of CCN (Mamouri and Ansmann, 2016; Choudhury and Tesche 2022). Most of the retrieved $N_{CCN}$ profiles are yet to be validated against in situ $N_{CCN}$ profile measurements."

Lines 100-104. The manuscript presents vertical distribution of Nccn for different regions within the NCP. Currently, we have satellite-based Nccn retrieval algorithms, for instance, Mamouri and Ansmann (2016) and Choudhury and Tesche (2022), to estimate profiles of Nccn from CALIPSO measurements. The in-situ measurements presented here will also be beneficial in validating such algorithms. This information is missing in the motivation.

**RE:** The sentence: "The in-situ measurements presented here are beneficial in validating lidar- or satellite-based $N_{CCN}$ retrieval algorithms (e.g., Choudhury and Tesche, 2022)." is added.

Lines 242-243: The Nccn values first increases till the base of the first temperature inversion layer (TIL). It is quite strange as the Nccn in the previous case with one TIL were more or less uniform below the layer, perhaps due to vertical mixing, which is not seen for this case with two inversion layers. Is there a possible reason behind this pattern?

**RE:** This is because the vertical mixing and the terrain effect make aerosol accumulate below the planetary boundary layer (PBL) on some days, which can be seen on the lidar image of a micro-pulse lidar (MPL) deployed during our field campaign at the Xingtai (XT) supersite. Unfortunately, the MPL data are missing during the RF1_1 on May 8, 2016. Figure S3h in the supplement suggests that $N_{CCN}$ also increases with height below the planetary boundary layer (PBL) during the RF8_1 on May 28, 2016. The MPL image shown below indicates that aerosol accumulates obviously in the upper PBL in the daytime. According to our measurements, this phenomenon has no relationship with TIL amount.

[Figure]

MPL image on May 28, 2016 at Xingtai (XT) supersite.

Table 1. As the flights measurements are taken in a spiral path, please mention the maximum horizontal distance covered by individual flight segments chosen in this study. This is important as you consider them as individual profiles later in the paper.

**RE:** The maximum spiral radius during every vertical spiral flight is added in Table 1. The updated table is shown in the below.

| Flight number, date | Time range (CST) | Flight code | Region covered | Vertical height a.s.l. (km) | Sampling duration (min) | Maximum spiral radius (km) |
|---|---|---|---|---|---|---|
| RF1, 20160508 | 13:02–14:29 | RF1_1 | XT | 0.3–3.7 | 38 | ~ 10 |
| | | RF1_a | track from XT to LC | ~3.6 | 20 | – |
| | | RF1_2 | LC | 0.3–3.2 | 15 | ~ 10 |
| RF2, 20160515 | 12:17–15:04 | RF2_a | track from LC to JL | ~0.4 | 18 | – |
| | | RF2_1 | JL | 0.3–3.6 | 40 | ~ 5.0 |
| | | RF2_2 | QZ | 0.3–3.6 | 38 | ~ 5.0 |
| | | RF2_b | track from QZ to JL | ~3.6 | 7 | – |
| | | RF2_c | track from JL to LC | ~0.4 | 10 | – |
| RF6, 20160521 | 12:04–14:41 | RF6_1 | QZ | 0.3–3.1 | 36 | ~ 5.0 |
| | | RF6_a | track from QZ to XT | ~2.5 | 18 | – |
| | | RF6_2 | XT | 0.3–2.6 | 43 | ~ 5.0 |
| | | RF6_b | track from XT to LC | ~1.1 | 13 | – |
| RF7, 20160528 | 10:21–13:25 | RF7_a | track around XT | ~3.1 | 20 | – |
| | | RF7_1 | XT | 0.5–3.1 | 49 | ~ 5.0 |
| | | RF7_b | track from XT to JL | ~0.4 | 10 | – |
| | | RF7_2 | JL | 0.3–2.5 | 26 | ~ 4.0 |
| | | RF7_c | track from JL to LC | ~1.8 | 7 | – |
| RF8, 20160528 | 16:30–18:24 | RF8_a | track around XT | ~0.6 | 15 | – |
| | | RF8_1 | XT | 0.5–3.1 | 36 | ~ 5.0 |
| RF11, 20160611 | 11:07–12:28 | RF11_a | track around XT | ~0.6 | 16 | – |
| | | RF11_1 | XT | 0.3–3.2 | 50 | ~ 4.0 |

Some important technical information are missing. Did you smooth the flight measurements before the analysis? The pre-processing done to the measurements should be discussed in Section 2. Please also provide the uncertainty or retrieval errors associated with the in-situ measurements.

**RE:** The CCNc data with instable sample or sheath flow are excluded. Considering the time reaching equilibrium at different $SS$ levels, data acquired in the final 30 s at any $SS$ level are used. The measurements of temperature ($T$) and potential temperature ($\theta$) are averaged in the intervals of 50 m in altitude. No other smoothing is applied. The uncertainty in nephelometer data is less than 10% (Anderson et al., 1996; Anderson and Ogren, 1998). The uncertain of effective water vapor supersaturation in CCNc is less than 5% (Rose et al., 2008).

The details of the flight plans, sampling method, and initial investigations into the impact of air mass on air chemistry have been published (Benish et al., 2020, 2021; F. Wang et al., 2018), and cited in our manuscript. It would be duplication if they were included in the main text, but we summarize them in the supplement.

**Reference:**

Anderson, T. L., Covert, D. S., Marshall, S. F., Laucks, M. L., Charlson, R. J., Waggoner, A. P., Ogren, J. A., Caldow, R., Holm, R. L., Quant, F. R., Sem, G. J., Wiedensohler, A., Ahlquist, N. A., and Bates, T. S.: Performance Characteristics of a High-Sensitivity, Three-Wavelength, Total Scatter/Backscatter Nephelometer, Journal of Atmospheric and Oceanic Technology, 13, 967-986, https://doi.org/10.1175/1520-0426(1996)013<0967:PCOAHS>2.0.CO;2, 1996.

Anderson, T. L., and Ogren, J. A.: Determining Aerosol Radiative Properties Using the TSI 3563 Integrating Nephelometer, Aerosol Science and Technology, 29, 57-69, https://doi.org/10.1080/02786829808965551, 1998.

Benish, S. E., He, H., Ren, X., Roberts, S. J., Salawitch, R. J., Li, Z., Wang, F., Wang, Y., Zhang, F., Shao, M., Lu, S., and Dickerson, R. R.: Measurement report: Aircraft observations of ozone, nitrogen oxides, and volatile organic compounds over Hebei Province, China, Atmospheric Chemistry and Physics, 20, 14523-14545, https://doi.org/10.5194/acp-20-14523-2020, 2020.

Benish, S. E., Salawitch, R. J., Ren, X., He, H., and Dickerson, R. R.: Airborne Observations of CFCs Over Hebei Province, China in Spring 2016, Journal of Geophysical Research: Atmospheres, 126, e2021J-e35152J, https://doi.org/10.1029/2021JD035152, 2021.

Rose, D., Gunthe, S. S., Mikhailov, E., Frank, G. P., Dusek, U., Andreae, M. O., and Pöschl, U.: Calibration and measurement uncertainties of a continuous-flow cloud condensation nuclei counter (DMT-CCNC): CCN activation of ammonium sulfate and sodium chloride aerosol particles in theory and experiment, Atmospheric Chemistry and Physics, 8, 1153-1179, https://doi.org/10.5194/acp-8-1153-2008, 2008.

Wang, F., Li, Z., Ren, X., Jiang, Q., He, H., Dickerson, R. R., Dong, X., and Lv, F.: Vertical distributions of aerosol optical properties during the spring 2016 ARIAs airborne campaign in the North China Plain, Atmospheric Chemistry and Physics, 18, 8995-9010, https://doi.org/10.5194/acp-18-8995-2018, 2018.

Lines 346-351: The definition and expression of scattering Ångström exponent should not be included in the "Results and Discussion" section. Please place it either in Section 2 or create a separate section.

**RE:** Agree. It is moved to Section 2.

Line 371: The section title is misleading. It is not the estimation of NCCN. It is where you parametrize NCCN in terms of aerosol optical properties. Please modify it.

**RE:** The section title is revised as: "Parametrizing $N_{CCN}$ in terms of aerosol optical properties".

Lines 379-383: Please refer Shinozuka et al. (2015) and correct the statements. Shinozuka et al. (2015) parameterize NCCN in terms of "extinction coefficient (at 500 nm)" and "Angstrom exponent" (calculated from extinction coefficients at 450 and 550 nm) for dry particles. They did not use scattering coefficient or scattering Angstrom exponent for the same. Also for equation 3, it should be stated that in Shinozuka et al. (2015), only the parameter β depends on the Angstrom exponent (computed from extinction coefficients).

**RE:** Shinozuka et al. (2015) identified $N_{CCN}$ at 0.4±0.1% *SS* with $10^{0.3\alpha}\sigma_{ext}^{0.75}$ where $\sigma_{ext}$ is the 500 nm extinction coefficient by dried particles and $\alpha$ is the extinction Angstrom exponent. They

determined the slope in $\log_{10}N_{CCN}$ vs. $\log_{10}\sigma_{ext}$ to be constant based on a variety of airborne and ground-based observations. However, we found that the slope varies with the extinction Angstrom exponent at some sites (such as the site of Black Forest, Germany) form Fig. 3a and Table 2 in Shinozuka et al. (2015). Therefore, we used the modified parameterization. The sentence is revised as: "Shinozuka et al. (2015) identified $N_{CCN}$ at 0.4±0.1% $SS$ with $10^{0.3\alpha}\sigma_{ext}^{0.75}$ where $\sigma_{ext}$ is the 500 nm extinction coefficient by dried particles and $\alpha$ is the extinction Angstrom exponent. According to our measurements, a modified parameterization is used in this study:

$$N_{CCN}=10^{\beta}\cdot\sigma^{\gamma} \tag{4}$$ ".

Lines 387-388: Coefficient of determination or R2 and correlation are synonymously used. R2 quantifies the goodness of fit (here linear fit) or performance of the model (here linear model) in simulating the variable of concern (here fitting parameters β and γ). I suggest using either correlation coefficient or slope of the linear fit. Also, is Figure 7 really important to include in the manuscript? I would suggest omitting the figure. If the authors want to retain it, they should justify the significance of the observed relations between SAE and the fitting parameters.

**RE:** Shinozuka et al. (2015) investigated the relationships of the slope ($\beta$) and intercept ($\gamma$) with the extinction Angstrom exponent ($\alpha$) shown in their Fig. 3. Following their methods, we also analyze these relationships in our study. The sentence is revised as: "The correlation coefficients ($R^2$) are lower in northwesterly air masses than in southeasterly air masses, likely due to more complex aerosol sources in northwesterly air masses." The figure is moved to the supplement.

Figure 7 (if retained) and Figure 8 should include the total number of points used in the comparison. I also suggest including two more lines in the figure representing one order of magnitude more and less than the 1:1 line in Figure 8 for better visualization.

**RE:** Agree. The updated figure is shown in the below.

[Figure]

**Figure 7.** Comparisons between measured $N_{CCN}$ at 0.7% $SS$ and estimated $N_{CCN}$ at 0.7% $SS$ using Eqs. (4) and (5) for different vertical spiral flights in (a) northwesterly and (b) southeasterly air masses. The black solid lines are 1:1 line and the dash lines indicate the boundaries representing ±10% deviations of $N_{CCN}$ (estimated) from $N_{CCN}$ (measured) in the log-log plot. The 10% deviation means that the deviation of individual data points is typically within a factor of 1.26 of the best

estimates. The point number ($N$) and root mean square error (RMSE) in each panel are given.

Lines 399-404: Qualitative interpretation from a log-log plot can be misleading. What seems to be different by a few millimeters in the plot can be different by orders of magnitude in reality. I suggest using parameters like normalized mean error or bias and root mean square error (normalized by mean) in percentage to get a better quantitative comparison. Such parameters should then be used to quantify the error associated with the proposed parametrization.

**RE:** The vertical variation of $N_{CCN}$ is in five orders of magnitude from a few to tens of thousands per cubic centimeter. Therefore, a log-log plot is commonly used. The root mean square error (RMSE) is calculated and shown in the above figure.

Is there any specific reason why the authors use aerosol scattering coefficient instead of extinction coefficient (scattering + absorption). The authors identify anthropogenic emissions as one of the aerosol types in their analysis, which may also include absorbing aerosols. Using scattering coefficient in such scenarios may result in mis-representation of absorbing aerosols in the parametrization, which is perhaps one of the reasons behind the errors in the predicted Nccn.

**RE:** It is true that absorbing aerosols can also serve as CCN, but their hygroscopicity is generally weak, such as black carbon (BC). In this study, we find that $N_{CCN}$ profile is impacted largely by anthropogenic emissions, especially air masses from the southeast. The impact of anthropogenic emissions does not only refer to primary processes. Our previous studies suggest that high concentration of gaseous precursors from anthropogenic emissions and strong atmospheric oxidization capacity lead to frequent new particle formation (NPF) and rapid particle growth in the NCP (Y. Wang et al., 2018, 2021; Zhang et al., 2018). These processes can produce many hydrophilic secondary aerosols (such as sulfate, nitrate, and so on), leading to the large increase of $N_{CCN}$. Moreover, the absorbing aerosols are much less than the scattering aerosols, which can be reflected from the value of single scattering albedo (SSA). F. Wang et al. (2018) reported that regional mean value of SSA at 550 nm in this campaign is 0.85±0.02.

The $N_{CCN}$ closure results using the data of extinction coefficients to estimate $N_{CCN}$ are shown below. The performance is similar with those using the data of scattering coefficients shown in the manuscript, indicating unimportant role of absorbing aerosols in the estimation of $N_{CCN}$.

[Figure]

[Figure]

**Reference:**

Wang, F., Li, Z., Ren, X., Jiang, Q., He, H., Dickerson, R. R., Dong, X., and Lv, F.: Vertical distributions of aerosol optical properties during the spring 2016 ARIAs airborne campaign in the North China Plain, Atmospheric Chemistry and Physics, 18, 8995-9010, https://doi.org/10.5194/acp-18-8995-2018, 2018.

Wang, Y., Li, Z., Zhang, Y., Du, W., Zhang, F., Tan, H., Xu, H., Fan, T., Jin, X., Fan, X., Dong, Z., Wang, Q., and Sun, Y.: Characterization of aerosol hygroscopicity, mixing state, and CCN activity at a suburban site in the central North China Plain, Atmospheric Chemistry and Physics, 18, 11739-11752, https://doi.org/10.5194/acp-18-11739-2018, 2018.

Wang, Y., Wang, J., Li, Z., Jin, X., Sun, Y., Cribb, M., Ren, R., Lv, M., Wang, Q., Gao, Y., Hu, R., Shang, Y., and Gong, W.: Contrasting aerosol growth potential in the northern and central-southern regions of the North China Plain: Implications for combating regional pollution, Atmospheric Environment, 267, 118723, https://doi.org/10.1016/j.atmosenv.2021.118723, 2021.

Zhang, Y., Du, W., Wang, Y., Wang, Q., Wang, H., Zheng, H., Zhang, F., Shi, H., Bian, Y., Han, Y., Fu, P., Canonaco, F., Prévôt, A. S. H., Zhu, T., Wang, P., Li, Z., and Sun, Y.: Aerosol chemistry and particle growth events at an urban downwind site in North China Plain, Atmospheric Chemistry and Physics, 18, 14637-14651, https://doi.org/10.5194/acp-18-14637-2018, 2018.

For identifying the aerosol types in the analyzed samples, HYSPLIT back trajectory analysis is used to track the source regions and the regions through which the air parcels have passes before reaching the target. However, this is based on the assumption that the lifetime of aerosols is long enough to retain its source identity. One of the ways to crosscheck the aerosol types is to use CALIPSO aerosol product (CALIPSO, 2018) for the identified cases. If there is no CALIPSO overpass over the region of interest at the desired time, one can also use re-analysis datasets like CAMS (Inness et al., 2019) and/or MERRA-2 (Molod et al., 2015) to identify the aerosol types that are dominant at different height levels. I suggest using either one of these datasets to check if the assumed aerosol signatures are correct.

**RE:** HYSPLIT has been used in a variety of simulations describing the atmospheric transport, dispersion, and deposition of pollutants. CALIPSO aerosol product is not suitable for our study chiefly because of lack of overpass over our observation site. We check the data of aerosol chemical composition from MERRA-2 (shown in the below). The results have a big difference with our measurement by an Aerodyne aerosol chemical speciation monitor (ACSM) at XT supersite (Zhang et al., 2018). The patterns of aerosol profiles from MERRA-2 are also not consistent with our measurements. On the contrary, our measurement data can be used to validate MERRA-2.

[Figure]

The vertical profiles of aerosol chemical composition from MERRA-2

**Reference:**

Zhang, Y., Du, W., Wang, Y., Wang, Q., Wang, H., Zheng, H., Zhang, F., Shi, H., Bian, Y., Han, Y., Fu, P., Canonaco, F., Prévôt, A. S. H., Zhu, T., Wang, P., Li, Z., and Sun, Y.: Aerosol chemistry and particle growth events at an urban downwind site in North China Plain, Atmospheric Chemistry and Physics, 18, 14637-14651, https://doi.org/10.5194/acp-18-14637-2018, 2018.

Minor comments:

Please modify Figure 1 caption to include the meaning of RF1, RF2… RF11.

**RE:** The sentence "The number after 'RF' indicates the research flight number" is added in the Fig. 1 caption.

Line 128. Please include the word in bracket. … 182 m above [mean] sea level …

**RE:** The word is added.

Line 295. Remove the word in the bracket. "…profiles [ss] are influenced…"

**RE:** The word is removed.

Lines 301-302. Rephrase the sentence to "Twomey (1959) first reported an exponential relationship between Nccn and ss."

**RE:** It is revised. Thanks.

In Figure 5, please mention that the y-axis is in log-scale. Please mark at least two (or three, if possible) tick labels in the y-axis of each plot.

**RE:** The sentence "The y-axis is logarithmic." is added in the Figure caption. The updated figure is shown in the below.

[Figure]

Line 345. The acronym "SAE" is previously defined in the paper. There is no need to define it again here.

**RE:** It is revised. Thanks.

Lines 349-351. Replace the word "dominated" by "are dominant".

**RE:** It is revised. Thanks.

References:

CALIPSO: Cloud-Aerosol Lidar and Infrared Pathfinder Satellite Observation Lidar Level 2 Aerosol Profile, V4-20, NASA Langley Atmospheric Science Data Center DAAC [data set], https://doi.org/10.5067/CALIOP/CALIPSO/LID_L2_05KMAPRO-STANDARD-V4-20, 2018.

Choudhury, G. and Tesche, M.: Estimating cloud condensation nuclei concentrations from CALIPSO lidar measurements, Atmos. Meas. Tech., 15, 639–654, https://doi.org/10.5194/amt-15-639-2022, 2022.

Inness, A., Ades, M., Agustí-Panareda, A., Barré, J., Benedictow, A., Blechschmidt, A.-M., Dominguez, J. J., Engelen, R., Eskes, H., Flemming, J., Huijnen, V., Jones, L., Kipling, Z., Massart, S., Parrington, M., Peuch, V.-H., Razinger, M., Remy, S., Schulz, M., and Suttie, M.: The CAMS reanalysis of atmospheric composition, Atmos. Chem. Phys., 19, 3515–3556, https://doi.org/10.5194/acp-19-3515-2019, 2019.

Mamouri, R.-E. and Ansmann, A.: Potential of polarization lidar to provide profiles of CCNand INP-relevant aerosol parameters, Atmos. Chem. Phys., 16, 5905–5931, https://doi.org/10.5194/acp-16-5905-2016, 2016.

Molod, A., Takacs, L., Suarez, M., and Bacmeister, J.: Development of the GEOS-5 atmospheric general circulation model: evolution from MERRA to MERRA2, Geosci. Model Dev., 8, 1339–1356, https://doi.org/10.5194/gmd-8-1339-2015, 2015.

---

## Author Comment (AC2)

**Reply to AC2**

We thank the reviewer for providing insightful comments and helpful suggestions that have substantially improved the manuscript. Below we have included the review comments in black followed by our responses in blue. In the revision of this manuscript, we have highlighted those changes accordingly.

The manuscript on "Vertical profiles of cloud condensation nuclei number concentration and its empirical estimate from aerosol optical properties over the North China Plain" by R. Zhang and co-authors made airborne measurements of vertical profiles of CCN concentrations and scattering coefficients over the southern plain of Hebei province. Using this data, they have investigated the influence of thermal structure (TIL) and airmass origin on vertical profiles of CCN. The CCN concentration is estimated using the scattering coefficient and its spectral variation.

Considering the limitations and uncertainties associated with the retrieval of vertical profiles of aerosols and CCN using different techniques, the direct measurements of these parameters onboard the aircraft are very important. But I feel disappointed with the way the authors described their experimental details. Details of the sampling inlet are not provided. (i) What is the effect of aircraft propeller on aerosol sampling? (ii) Whether sampling flow was iso-kinetic? (iii) What was the sampling efficiency of the inlet used? (iv) What was the cruising speed of the Y-12 Turboprop? (v) How do authors account for ram heating? (vi) How do authors account for the flow instabilities during ascending and descending phases of spiral flights? (vii) How much is the total sampling time available for each vertical level? (viii) Whether CCN measurements at all the supersaturations were carried out at each altitude? If not, how do authors decouple the change in CCN due to supersaturation change and also due to vertical variation?

RE: Thanks for the suggestion. The details of the flight plans, sampling method, and initial investigations into the impact of air mass on air chemistry have been published (Benish et al., 2020, 2021; F. Wang et al., 2018), and cited in our manuscript. It would be duplication if they were included in the main text, but we summarize them here below for the sake of the reviewer and we have included the description in the supplement.

(i): The sampling device (shown in the figures below) is above the front of the airplane cabin, which is not affected by the propeller after the plane takes off.

[Figure]

(ii) and (iii): The sampling flow was iso-kinetic. As described in F. Wang et al. (2018), the conical double diffuser aerosol inlet, designed for a Twin Otter, was installed on the Y-12. This inlet system was manufactured by Droplet Measurements Technologies (MP-1806-A and MP-1807-A, Boulder, CO, USA) (Hegg et al., 2005). It has been used extensively on the University of Maryland's Cessna 402 (Brent et al., 2015). The passing efficiency is expected to be near 100% for particle diameters up to 2.5 μm and near 50% for particles between 3 and 4 μm (Huebert et al., 2004; McNaughton et al., 2007).

(iv): As described in F. Wang et al. (2018), the typical cruising speed of aircraft is 60-70 m s$^{-1}$, with ascent/descent rates of 2–5 m s$^{-1}$.

(v): Ascents and descents were gentle to avoid turbulence taking about 20 min to ascend 3000 m or ~150 m/min. The ram heating was considered by adjusting the measured air temperature and relative humidity:

$$Temp\_adj = (Temp + 273.15) / (1 + 0.2 * rf * M^2) - 273.15;$$

Where,

Temp_adj – adjusted air temperature by taking the ram heating effect into account

Temp – measured air temperature (°C)

rf – recovery factor (rf = 0.896445604404384)

M – mach number, which is calculated from the measured true air speed and calculated speed of sound:

$$M = Airspeed\_True / Speed\_sound$$
$$Speed\_sound = 331.3 * sqrt ((Temp + 273.15) /273.15)$$

Relative humidity was also adjusted by multiplying the ratio of saturated water pressures under measured and adjusted air temperature:

$$RH\_adj=RH. *(svpt./svpat);$$

where,

svpt=6.1121 * exp((18.678 – Temp / 234.5) * (Temp / (257.14 + Temp)));

svpat=6.1121 * exp ((18.678 - Temp_adj / 234.5) * (Temp_adj / (257.14 + Temp_adj)));

(vi): When not on a smooth ascent or descent the sampling time at each level varied from ~2 to ~20 min.

(vii): The sampling duration of every vertical spiral or level flight is added in the updated Table 1 shown in the below.

| Flight number, date | Time range (CST) | Flight code | Region covered | Vertical height a.s.l. (km) | Sampling duration (min) | Maximum spiral radius (km) |
|---|---|---|---|---|---|---|
| RF1, 20160508 | 13:02–14:29 | RF1_1 | XT | 0.3–3.7 | 38 | ~ 10 |
| | | RF1_a | track from XT to LC | ~3.6 | 20 | – |
| | | RF1_2 | LC | 0.3–3.2 | 15 | ~ 10 |
| RF2, 20160515 | 12:17–15:04 | RF2_a | track from LC to JL | ~0.4 | 18 | – |
| | | RF2_1 | JL | 0.3–3.6 | 40 | ~ 5.0 |
| | | RF2_2 | QZ | 0.3–3.6 | 38 | ~ 5.0 |
| | | RF2_b | track from QZ to JL | ~3.6 | 7 | – |
| | | RF2_c | track from JL to LC | ~0.4 | 10 | – |

| | | | | | | |
|---|---|---|---|---|---|---|
| | | RF6_1 | QZ | 0.3–3.1 | 36 | ~ 5.0 |
| RF6, 20160521 | 12:04– 14:41 | RF6_a | track from QZ to XT | ~2.5 | 18 | – |
| | | RF6_2 | XT | 0.3–2.6 | 43 | ~ 5.0 |
| | | RF6_b | track from XT to LC | ~1.1 | 13 | – |
| RF7, 20160528 | 10:21– 13:25 | RF7_a | track around XT | ~3.1 | 20 | – |
| | | RF7_1 | XT | 0.5–3.1 | 49 | ~ 5.0 |
| | | RF7_b | track from XT to JL | ~0.4 | 10 | – |
| | | RF7_2 | JL | 0.3–2.5 | 26 | ~ 4.0 |
| | | RF7_c | track from JL to LC | ~1.8 | 7 | – |
| RF8, 20160528 | 16:30– 18:24 | RF8_a | track around XT | ~0.6 | 15 | – |
| | | RF8_1 | XT | 0.5–3.1 | 36 | ~ 5.0 |
| RF11, 20160611 | 11:07– 12:28 | RF11_a | track around XT | ~0.6 | 16 | – |
| | | RF11_1 | XT | 0.3–3.2 | 50 | ~ 4.0 |

(viii): As described in section 2.2, CCNc-200 has two columns that can simultaneously measure $N_{CCN}$ at two different supersaturation ($SS$) levels without mutual interference. In this campaign, only one $SS$ level (0.7%) was set in the first column during all flights, but eight different $SS$ levels (0.44%, 0.56%, 0.68%, 0.80%, 0.92%, 1.04%, 1.16%, and 1.28%) were set in the second column with a measurement time interval of 90 s for each $SS$ level. We can get continuous $N_{CCN}$ data at 0.7% $SS$ but not at other $SS$ during any flight. However, $N_{CCN}$ data with different $SS$ at a certain altitude can be obtained during the level flights. Therefore, we can analyze the vertical profile of $N_{CCN}$ with 0.7% $SS$ (Fig. 3–4) and CCN spectra using $N_{CCN}$ data with different $SS$ at certain altitudes (Fig. 5).

**Reference:**

Benish, S. E., He, H., Ren, X., Roberts, S. J., Salawitch, R. J., Li, Z., Wang, F., Wang, Y., Zhang, F., Shao, M., Lu, S., and Dickerson, R. R.: Measurement report: Aircraft observations of ozone, nitrogen oxides, and volatile organic compounds over Hebei Province, China, Atmospheric Chemistry and Physics, 20, 14523-14545, https://doi.org/10.5194/acp-20-14523-2020, 2020.

Benish, S. E., Salawitch, R. J., Ren, X., He, H., and Dickerson, R. R.: Airborne Observations of CFCs Over Hebei Province, China in Spring 2016, Journal of Geophysical Research: Atmospheres, 126, e2021J-e35152J, https://doi.org/10.1029/2021JD035152, 2021.

Brent, L., Thorn, W., Gupta, M., Leen, B., Stehr, J., He, H., Arkinson, H.,Weinheimer, A., Garland, C., and Pusede, S.: Evaluation of the use of a commercially available cavity ringdown absorption spectrometer for measuring NO2 in flight, and observations over the Mid-Atlantic States, during DISCOVER-AQ, Journal of Atmospheric Chemistry, 72, 503–521, https://doi.org/ 10.1007/s10874-013-9265-6, 2015.

Hegg, D. A., Covert, D. S., Jonsson, H., and Covert, P. A.: Determination of the transmission efficiency of an aircraft aerosol inlet, Aerosol Science and Technology, 39, 966–971, https://doi.org/10.1080/02786820500377814, 2005.

Huebert, B., Bertram, T., Kline, J., Howell, S., Eatough, D., and Blomquist, B.: Measurements of organic and elemental carbon in Asian outflow during ACE-Asia from the NSF/NCAR C-130, Journal of Geophysical Research: Atmospheres, 109, D19S11, https://doi.org/10.1029/2004JD004700, 2004.

McNaughton, C. S., Clarke, A. D., Howell, S. G., Pinkerton, M., Anderson, B., Thornhill, L.,

Hudgins, C., Winstead, E., Dibb, J. E., and Scheuer, E.: Results from the DC-8 Inlet Characterization Experiment (DICE): Airborne versus surface sampling of mineral dust and sea salt aerosols, Aerosol Science and Technology, 41, 136–159, https://doi.org/10.1080/02786820601118406, 2007.

Wang, F., Li, Z., Ren, X., Jiang, Q., He, H., Dickerson, R. R., Dong, X., and Lv, F.: Vertical distributions of aerosol optical properties during the spring 2016 ARIAs airborne campaign in the North China Plain, Atmospheric Chemistry and Physics, 18, 8995-9010, https://doi.org/10.5194/acp-18-8995-2018, 2018.

The authors mentioned that CCN profiles have a strong dependence on the number and thickness of TIL. This is mostly due to the TIL influence on the vertical transport of aerosols. On the other hand, the influence of airmass trajectory indicates long-range transport. In other words, when long-range transport dominates at higher altitudes, the influence of vertical transport of aerosols from the lower atmosphere is irrelevant. If long range transport is the prominent mechanism, then how could authors associate TIL with CCN concentration?

RE: The $N_{CCN}$ profiles (Fig. 4) differ significantly for airmasses from the northwest vs southeast. In air masses from the northwest, air from the western desert or plateau carries a large number of aerosols to the NCP. Aerosols do not accumulate near the surface due to the rapid dispersion. Therefore, the TIL effect on $N_{CCN}$ profiles is weak under the impact of air masses from the northwest. This is why the vertical variation of $N_{CCN}$ is small in northwesterly air masses (Fig. 4b). In southeasterly air masses, aerosols near the surface accumulate easily due to the terrain blocking effect by the Taihang Mountains, leading to the strong vertical transport of aerosols. At the same time, the long-range transport of aerosols from northwest is weak. Thus, much lower $N_{CCN}$ above 2 km than near the surface. The effect of long-range transport is dominant in northwesterly air masses, while the effect of vertical transport (the effect of TIL) is dominant in southeasterly air masses. The two effects are caused by the different meteorological conditions and special terrain distribution.

The long-range transports of aerosols play an important role in the structure of $N_{CCN}$ profiles. However, the role of TIL cannot be ignored. In this campaign, a micro-pulse lidar (MPL) was deployed at the Xingtai (XT) supersite. On some days, there are different aerosol layers, which can be reflected by the MPL images. Some examples are shown below (green colors in the lower atmosphere indicate aerosol layers). Aerosol vertical stratification should be related to the influence of TILs.

[Figure]

[Figure]

MPL images on different days

How do authors link CCN spectra with activation efficiency? In lines 313-314, the authors mentioned that "A lower value means a stronger aerosol activation ability (i.e., more coarse-mode particles or stronger aerosol hygroscopicity), and vice versa." This is not always true when hygroscopicity changes with the size of the particles.

RE: According to Köhler theory, aerosol hygroscopicity or activation ability is controlled by the Raoult effect and the Kelvin effect. An increase in particle size can enhance aerosol hygroscopicity or activation ability due to the Kelvin effect. Many studies used the $k$ parameter to analyze aerosol activation ability. For example, Jefferson (2010) suggested that the $k$ parameter indicates the steepness of the change in CCN concentration with $SS$. Low values of $k$ are typical of highly soluble aerosol such as sea salt and high $k$ values of low-solubility aerosols.

**Reference:**

Jefferson, A.: Empirical estimates of CCN from aerosol optical properties at four remote sites, Atmospheric Chemistry and Physics, 10, 6855-6861, https://doi.org/10.5194/acp-10-6855-2010, 2010.

How much time CCN counter required for attaining set supersaturation, especially when supersaturation changes from 1.28% to 0.44%? What is the sanctity of 0.7% supersaturation? Why lower supersaturations (<0.4%) are excluded from the sampling? What is the broad range of atmospheric supersaturation observed over the study region?

RE: In this campaign, the supersaturation of CCNc is adjusted by the control of flow rate (Rose et al., 2008). Therefore, the supersaturation adjustment for any change is rapid. Eight different *SS* levels are set in the second column with an observation interval of 90 s. Considering the equilibrium time of *SS* levels, data from the final 30 s data at any *SS* level in the cycle for the second column are used in this study. We focus on the impact of aerosols on the convective clouds. Therefore, the set supersaturation is high. In addition, the low $N_{CCN}$ value at low *SS* in the free troposphere approaching CCNc measurement limit can make a large uncertainty.

**Reference:**

Rose, D., Gunthe, S. S., Mikhailov, E., Frank, G. P., Dusek, U., Andreae, M. O., and Pöschl, U.: Calibration and measurement uncertainties of a continuous-flow cloud condensation nuclei counter (DMT-CCNC): CCN activation of ammonium sulfate and sodium chloride aerosol particles in theory and experiment, Atmospheric Chemistry and Physics, 8, 1153-1179, https://doi.org/10.5194/acp-8-1153-2008, 2008.

What kind of drier was used to remove the humidity of the air sampled by the nephelometer? Whether this could maintain a constant RH throughout the campaign?

RE: We did not dry the air sampled by the Nephelometer. Instead, we adjust for increased scattering with increased relative humidity with a correction factor, $f(RH)$, which is calculated by:

$$f(RH) = \left[\frac{(100 - RH_{neph})}{(100 - RH_{amb})}\right]^{\gamma} \qquad B_{scat\_adj} = B_{scat} \times C \times f(RH)$$

Where,

$RH_{neph}$ – Internal Nephelometer RH

$RH_{amb}$ – Adjusted Ambient RH that takes the ram heating into account (see above for the adjustment)

$B_{scat\_adj}$ – Adjusted Scattering Coefficient

$B_{scat}$ – Measured Scattering Coefficient

$\gamma$ – Measured Dry vs Humid Factor

C – Angular Truncation Factor, which is empirically derived based on the method by Anderson and Ogren (1998):

| λ | Angular Truncation Factor | Angstrom Exponent | Detection Limit |
|---|---|---|---|
| **450 nm** | $C^{450} = 1.165 - 0.046 \times A_{500}$ | $A_{500} = -\log(b_{scat}^{450}/b_{scat}^{550})/\log(450/550)$ | 4.4E-07 m$^{-1}$ |
| **550 nm** | $C^{550} = 1.152 - 0.044 \times A_{575}$ | $A_{575} = -\log(b_{scat}^{450}/b_{scat}^{700})/\log(450/700)$ | 1.7E-07 m$^{-1}$ |
| **700 nm** | $C^{700} = 1.120 - 0.035 \times A_{625}$ | $A_{625} = -\log(b_{scat}^{550}/b_{scat}^{700})/\log(550/700)$ | 2.6E-07 m$^{-1}$ |

**Reference:**

Anderson, T. L. and Ogren, J. A.: Determining aerosol radiative properties using the TSI 3563 integrating Nephelometer, Aerosol Science and Technology, 29, 57–69, https://doi.org/10.1080/02786829808965551, 1998.

Line 173: Replace "this" with "integrating"

RE: Revised. Thanks.

There are data gaps in Figure 5. For example (i) panel a RF2_c: no CCN data is shown for ss<0.8%.

Similar is the case with panel b RF6_b and panel C RF7_c. Explain?

RE: The sampling duration for every vertical spiral or level flight is added in the updated Table 1 shown in the reply to first comment. In some level flights of short sampling durations, we couldn't obtain $N_{CCN}$ data at all $SS$. Even so, there are sufficient data points in Fig. 5 for fitting.

What is the reason for high CCN activation at higher altitudes than lower levels? Normally, fine mode aerosols are transported to higher altitudes and these particles have lower CCN efficiency than coarse mode aerosols.

RE: What is stated by the reviewer may be generally true, but it varies from case to case. Scattering Ångström exponent (SAE) is often used to qualitatively assess the dominant size mode of aerosol activation, reflecting the particle number size distribution (PNSD) pattern (e.g., Hamonou et al., 1999). A large SAE (> 2) generally implies that fine-mode aerosols are dominant (e.g., smoke particles), while a small SAE (< 1) means that coarse-mode aerosols are dominant (e.g., dust particles). The SAE profiles shown in Fig. 6a suggests that the coarse mode particles are dominant above 2 km, while fine-mode aerosols are dominant below 2 km. The origin of the air shifts generally to the northwest with increasing altitude and the origin of the aerosols changes with it. In addition, particles can age and grow during the transport processes due to the atmospheric chemical reactions such as cloud processing. Liu et al., (2019) indicated that the mass fraction of hydrophilic secondary aerosols is higher in the upper atmosphere than near the surface based on the in-situ aircraft measurements in Beijing in the NCP.

**Reference:**

Hamonou, E., Chazette, P., Balis, D., Dulac, F., Schneider, X., Galani, E., Ancellet, G., and Papayannis, A.: Characterization of the vertical structure of Saharan dust export to the Mediterranean basin, Journal of Geophysical Research: Atmospheres, 104, 22,257-22,270, https://doi.org/10.1029/1999jd900257, 1999.

Liu, Q., Quan, J., Jia, X., Sun, Z., Li, X., Gao, Y., and Liu, Y.: Vertical Profiles of Aerosol Composition over Beijing, China: Analysis of In Situ Aircraft Measurements, Journal of the Atmospheric Sciences, 76, 231-245, https://doi.org/10.1175/JAS-D-18-0157.1, 2019.

How does long-range transport increase SAE? Generally, ageing and chemical processing during the long-range transport increases the size of the particles and reduces SAE. Moreover, ultrafine secondary particles have less residence time and they may not get transported to longer distances to increase SAE.

RE: Agree. In section 3.3.1, we indicate that the long-distance transport of coarse-mode aerosols (like dust particles) decreases SAE in the free troposphere.

Better association between CCN at high $SS$ and scattering coefficients are expected because both CCN and scattering coefficients depend on the entire size distribution of the aerosol system. On the other hand, predicting CCN concentration for lower $SS$ is challenging, since a small portion of the aerosol NSD (coarse mode) gets activated. Using the high-resolution data (1 sec), the authors should show the CCN vs scattering coefficients for low and high supersaturations.

RE: Agree. In the campaign, $N_{CCN}$ at 0.7% $SS$ is measured by a separate column of CCNc but not for other $SS$s. Therefore, the data of $N_{CCN}$ at 0.7% $SS$ is enough to do the closure study. However,

the samples of $N_{CCN}$ at other $SS$ are too few to obtain a meaningful closure result. The figure below depicts $N_{CCN}$ closure test at other $SS$. Overall, the closure performance is similar to that at 0.7% $SS$ but the number of data points at any $SS$ is low.

[Figure]

Figure 7: Standard deviation of the β and γ should be included.

RE: Agree. The updated figure is shown in the below. The figure has been removed in the supplement.

[Figure]

The β and γ showed better association with SAE during the southeast airmass period than the

northwest airmass. But the CCN estimated using b and g did not show good association for southeast airmass. Please explain this discrepancy.

RE: The data from two vertical spiral flights (RF1_1 and RF1_2) in northwesterly air masses are used to do the $N_{\text{CCN}}$ closure work, while the data from five spiral flights (RF6_1, RF6_2, RF7_1, RF7_2, and RF8_1) in southeasterly air masses are used. More data in southeasterly air masses can be used, leading to better fitting for $\beta$ and $\gamma$. However, the larger difference of $N_{\text{CCN}}$ values between five spiral flights in southeasterly air masses worsen the closure performance. More aircraft data will be needed to establish a more reasonable parameterization scheme for $N_{\text{CCN}}$ at different $SS$ in the NCP.